# Native and Indigenous Populations and Gastric Cancer: A Worldwide Review

**DOI:** 10.3390/ijerph19095437

**Published:** 2022-04-29

**Authors:** Felina M. Cordova-Marks, William O. Carson, Angela Monetathchi, Alyssa Little, Jennifer Erdrich

**Affiliations:** 1Department of Health Promotion Sciences, Zuckerman College of Public Health, University of Arizona, Tucson, AZ 85724, USA; williamocarson@email.arizona.edu; 2Cellular Molecular Medicine, College of Medicine, University of Arizona, Tucson, AZ 85724, USA; amonetathchi@email.arizona.edu; 3University of Arizona, Tucson, AZ 85724, USA; alyssalittle@email.arizona.edu; 4Department of Surgery, College of Medicine, University of Arizona, Tucson, AZ 85724, USA; jerdrich@surgery.arizona.edu

**Keywords:** gastric cancer, American Indians, structural racism, health disparities

## Abstract

Gastric cancer is a worldwide concern, particularly for Indigenous populations who face greater disparities in healthcare. With decreased access to screening and critical treatment delays, this group is experiencing adverse health effects. To determine what factors drive these disparities, a systematic review was performed in PubMed. This revealed a lack of research on gastric cancer specific to this population. The literature primarily focused on subset analyses and biological aspects with sparse focus on determinants of health. The results informed this presentation on factors related to Indigenous gastric cancer, which are influenced by colonialism. Indigenous populations encounter high rates of food shortage, exposure to harmful environmental agents, structural racism in the built environment, *H. pylori*, and compromised healthcare quality as an effect of colonialism, which all contribute to the gastric cancer burden. Putting gastric cancer into a cultural context is a potential means to respond to colonial perspectives and their negative impact on Indigenous patients. The objective of this manuscript is to examine the current state of gastric cancer literature from a global perspective, describe what is currently known based on this literature review, supplemented with additional resources due to lack of published works in PubMed, and to present a model of gastric cancer through the lens of a modified medicine wheel as a potential tool to counter colonial healthcare perspectives and to honor Indigenous culture.

## 1. Introduction

### Incidence and Prevalence of Gastric Cancer: A Worldwide Perspective

The international burden of gastric cancer is startling as it is the sixth most common cancer that has resulted in the fourth highest cancer-related mortality worldwide in 2020 [1]. In the United States, the burden is not as steep, as gastric cancer is the 15th most common cancer, with incidence and mortality rates trending downwards, and an upward trend in 5-year survival rates [2]. Gastric cancer is more common in middle-aged and older populations, with the average age of diagnosis at 68 years [2]. Males have higher incidence (9.7 vs. 5.3) and mortality (3.9 vs. 2.1) rates per 100,000 than females [2]. Race/ethnicities other than Non-Hispanic White have the highest rates; Black males, and Hispanic females have both the highest incidence and mortality rates in the United States [2].

Although gastric cancer has decreased globally over the last several decades, incidence and mortality rates remain elevated for Indigenous people [3]. In Taiwan, the Indigenous population has nearly twice the incidence and more than double the mortality rate compared to their non-Indigenous counterparts [4]. In New Zealand, the Indigenous Maori population has experienced an increased incidence and higher mortality rate for gastric cancer compared to non-Maori [5]. Indigenous South Americans also report high rates of gastric cancer [6]. In the United States, American Indians/Alaska Natives have an incidence rate of 14.2/100,000 for males, and 7.6/100,000 for females compared to an incidence rate of 7.6/100,000 for White males and 3.3/100,000 for White females in the same regions [3]. Mortality rates tell a similar story with American Indians/Alaska Natives having a mortality rate of 6.9/100,000 for males and 3.3/100,000 for females [2] compared to mortality rates of 2.8/100,000 for males and 1.5/100,000 for females among the White population [2]. In addition, incidence rates have been on the rise among Alaska Natives, younger populations and females [7]. Although American Indians and Alaska Natives do not have the highest incidence or mortality rate at the national level, these rates can vary greatly by geographic location. In Northern Arizona, the rates for gastric cancer are approximately three times higher among American Indians than the general Arizona population, and it is also the cancer type with the highest mortality rate among American Indians [8]. Indigenous people are more likely to have lower access to screening and are diagnosed at later stages [3]. Diagnosis of advanced-stage gastric cancer occurs at an earlier age among Alaska Native people [7]. SEER data from 2007 to 2015 demonstrate that the median time of survival from diagnosis is 23 months among American Indians and Alaska Natives, compared to a median survival time of 40 months for the White population during the same period [9].

Gastric cancer incidence and mortality rates among Indigenous populations points to a need for increased research. Elucidation of health determinants with an emphasis on environmental factors, other ways of making a positive impact, and explanation through a cultural lens are needed. Arnold et al. published a systematic review of Indigenous gastric cancer worldwide, focusing on results up to 2013 and on incidence, mortality and survival [10]. We will focus on more recent publications, with the goal of looking at determinants of health and qualitative findings instead of rates, as this represents a major gap in the literature. The objectives of this manuscript are to: 1. Examine the current state (2011–2021) of gastric cancer globally and describe what is currently known based on this literature review 2. Supplement findings with additional resources to present a full picture of Indigenous gastric cancer; and 3. Present a culturally responsive model of gastric cancer as a potential tool to counter colonial healthcare perspectives and to honor Indigenous culture.

## 2. Methods

### 2.1. Systematic Literature Review

PRISMA was utilized to guide a systematic literature review which was conducted in November 2021 in PubMed with the search terms: (“gastric cancer” OR “stomach cancer”) AND (“Alaska Native” OR “Indigenous” OR “First Nation” OR “American Indian” OR “Aboriginal” OR “Maori” OR “Native American”) with a time constraint of publication within the past ten years, and only manuscripts published in English and full text articles included (Figure 1). We also looked at the difference in the number of publications in the past ten years versus the past five years to determine if there has been increased research more recently on this topic. Exclusion criteria included articles that were not full text, published before 2011, not inclusive of Indigenous populations, not related to gastric cancer, or only related to biology of gastric cancer (Figure 1).

### 2.2. Sources beyond the Systematic Review

Given the shallow pool of research on gastric cancer in Indigenous populations, the research team utilized additional sources that examined common risk factors for gastric cancer, via scientific research journals, government websites, and programming that highlights work by Indigenous people to alleviate risk factors for gastric cancer. We conducted a hand search of PubMed and Google Scholar using the prior search terms, as well as specific searches for the most common risk factors for gastric cancer.

## 3. Results of Review and Outside Sources Supplementary Findings

### 3.1. Systematic Literature Review

Over one hundred published articles were found in the literature review (Figure 1). We chose to limit searches to a ten-year period, leaving us with sixty total publications on PubMed. This number was reduced to 43 publications when the time frame was constricted to the last five years. This shows that 71.7% of all articles were published within the past five years, a marked increase for 2016–2021 in comparison to findings prior to 2016.

When all 60 of the studies from 2011–2021 were examined to see which matched our criteria focusing on Indigenous populations and gastric cancer research, 35 articles were found to fit. Of these articles, 12 (34.2%) were conducted with populations in the United States, with seven (20.0%) that looked at Alaska Native Populations. Canada and New Zealand researchers published seven articles (20.0%) on their Indigenous populations. After this, only Taiwan had more than a single article published on gastric cancer in their Indigenous populations, with two articles (5.7%).

Looking at the type of research produced worldwide in the 35 articles, the majority looked at aspects of gastric cancer biology. Of the 35 articles, 31 (88.5%) included biological function research, while only 14 (40.0%) looked at sociological constructs for elevated rates of gastric cancer. Twenty-one (60.0%) included secondary data analysis pieces, using surveillance data as the basis of their papers. Six of the articles (17.1%) were new quantitative studies conducted by researchers that did not utilize already collected data. Only one (2.8%) paper utilized community based participatory research and there was one separate single qualitative study (2.8%). The two papers classified as “Other” were an editorial and a summary article.

This review highlights a promising increase in research looking at gastric cancer among Indigenous populations, but also highlights a lack of variety in research and in the geography of where the research is being conducted. Most contemporary research on gastric cancer features statistics on the elevated rates with few focused on why gastric cancer is higher amongst Indigenous people worldwide, and how this can be reduced, particularly within a culturally responsive framework.

### 3.2. Sources beyond the Systematic Review

In total, we identified 54 articles to supplement the findings: six from fact sheets and Indigenous led programming efforts (found on websites), and an additional 48 papers on risk factors for gastric cancer for Indigenous people specifically. These articles from outside our systematic review help provide evidence for the specific challenges that Indigenous people face when addressing common risk factors for gastric cancer.

## 4. Literature Review Qualitative Findings

The systematic literature review points to an overall lack of published research, lack of variety in research, and the need for additional Indigenous-focused gastric cancer research. In the following sections of this manuscript, we utilize the publications found during this literature review (Figure 1) and the additional sources outside of this review/PubMed, described in Section 3.2. These sources were utilized to describe structural inequalities that uniquely contribute to gastric cancer among Indigenous populations. These inequalities include the following categories: food (access and choices), environmental concerns (uranium exposure, water contamination, infectious disease), and the health care system (inequitable access to cancer screening and treatment services, and cultural barriers). We also offer an explanation of gastric cancer through an Indigenous cultural lens that can potentially be utilized by both providers and Indigenous gastric cancer patients.

## 5. Food and Gastric Cancer

### 5.1. Access

Diets with excessive levels of salt, low fruit/vegetable content, and increased alcohol consumption increase the risk of gastric cancer [11]. Throughout the history of colonialism, tribal lands have been seized and populations removed onto land not conducive to farming or sustainable food sources [12]. Indigenous populations throughout the world have been stripped of their resources with many Indigenous people living in areas that are classified as ‘food deserts’ [13]. Indigenous populations are more likely to be food insecure compared to the general population of their country [14,15,16]. Urban Indigenous people face even greater food insecurity compared to rural Indigenous populations [15]. Traditional Indigenous food sources have also been replaced with less nutritious and highly processed foods [13]. Diets high in salt, red meat, and smoked foods have increased, whereas access to traditional foods and honoring of traditional food systems have declined due to socioeconomic, environmental, and political factors such as colonialism, the commodity foods program, increased cost of healthier food choices, and climate change [17].

### 5.2. Choices

Diet and lifestyle modification such as increased consumption of fruits/vegetables and reduced consumption of processed foods is routinely recommended to reduce the risk of gastric cancer [11], but many Indigenous people face barriers to these behavioral changes. Research from Australia shows that Indigenous populations consume a greater proportion of sugar-sweetened food and beverages compared to other populations in Australia [18]. A primary driver is the high price and decreased access to healthier food options and safe drinking water [18]. In Canada, a review of obesity among First Nations communities observed that fluctuating food prices by season, lack of access to healthy food options, and ease of access to convenience stores are factors in the increased levels of obesity [12,19]. A past review examining food access for Indigenous people globally highlighted that, for rural and remote indigenous peoples in Australia, Canada, Greenland, Guam, and the United States, often stores do not carry fresh items such as produce, dairy, and meat [16]. In the United States, according to the recent Tribal Resilience in Vulnerable Environments (THRIVE) study conducted with American Indian participants, only half reported that they had access to fresh fruits and vegetables at their local stores, with only a third reporting that the produce was of good quality [20]. In addition, THRIVE found that more than half of the participants purchased groceries at gas stations, convenience stores, and dollar stores, locations that often provide mostly preserved and processed foods [20]. Distance is a barrier to accessing fresh food as more than half of these same participants also reported traveling over 20 miles to do grocery shopping [20].

## 6. Environmental Factors

Environmental determinants of health contribute to a broad spectrum of poor health outcomes, gastric cancer included [21]. Environmental violence can be seen in water contamination with uranium, documented as a major environmental risk factor for the development of gastric cancer [22].

### 6.1. Uranium Exposure

Environmental violence is a threat that Indigenous populations face, a prime example being uranium mining by non-Native/Indigenous companies, frequently conducted near or on Indigenous land. This has contaminated water sources, the soil, and wildlife that community members depend on for survival [21]. Absorption of uranium in the gastrointestinal tract and chronic environmental exposure to uranium increases the risk of gastric cancer and other malignancies [21]. In Australia, Aboriginal populations near the Ranger uranium mine have suffered elevated rates of cancer, compared to the general population, from the contaminated water supply [23]. Researchers in Bavaria, Germany, and South Carolina both documented that uranium exposure in drinking water may be related to increases in several types of cancer in the region [24,25]. Water in the Western United States has elevated levels of uranium, with measurements up to double the Environmental Protection Agency approved rate in areas that were once/currently near mining sites [23,26,27]. In the United States, many Tribal Nations including the Navajo Nation reside near uranium mines, with an estimated 40% of the western watershed contaminated with uranium [21]. Navajo (Diné) people living on or near uranium mines or working for coal companies and the railroad industry are significantly exposed to environmental contaminants [28]. The past infringement of mining for natural resources on Indigenous people’s lands may have produced unexpected negative consequences. In addition, prior research has shown an association between uranium mine locations and exposure to radiation, even at low levels, and risk of developing gastric cancer [22]. If we look to other populations, longitudinal studies conducted with German uranium miners showed increased risk for development of stomach cancer among those who were exposed to radiation [24].

### 6.2. The Built Environment and Water

The effects of colonialism and environmental racism have placed Indigenous peoples in areas that lack infrastructure for plumbing, electricity, refrigeration, and access to safe drinking water [29]. As previously mentioned, people may be situated in areas where their water sources are contaminated by outside mines and radiation [21,23,26,27]. Indigenous communities such as the Hopi Tribe in Arizona [30] and the Crow Tribe in Montana are situated in regions with difficult access to safe drinking water [31]. This is not an isolated problem as water security has long been discussed as an issue among global Indigenous populations [32,33,34,35]. The lack of access to in-home plumbing leads to increased risk of infectious disease, and less water consumption overall, propagating intake of sugar sweetened beverages and other unhealthy food choices [36]. Due to lack of refrigeration and running water, Navajo households with a lower socioeconomic status are hard-pressed to adopt the kind of lifestyle behaviors that reduce the risk of gastric cancer [28].

### 6.3. H. pylori

Research has shown that access to modern water services and plumbing decreases the risk of developing gastrointestinal infections such as helicobacter pylori (*H. pylori*), [37]. Alaska Native gastric cancer patients who live in rural areas frequently live in multi-generational homes that are often crowded and lack running water, increasing transmission of communicable diseases, *H. pylori* included [7]. Indigenous people are more likely to be infected with *H. pylori,* elevating risk for gastric cancer [3]. In Taiwan, Indigenous populations have a 27.9% higher prevalence of *H. pylori* than non-Indigenous [4]. Similar findings have been found in Maori of New Zealand, with Maori children having 14% higher prevalence of *H. pylori* infection and adults having a 21% higher prevalence than non-Maori [38]. In a prevalence study of Arctic Indigenous populations, *H. pylori* was found in 66% of participants [39]. In an older study, 92% of Indigenous populations of South America were found to be positive for *H. pylori* [40]. In the United States, similar prevalence is seen among American Indian populations, with 55% of Plains American Indians, 65.3% of Navajo participants in a pilot study, and 72% of Navajo patients at a healthcare facility testing positive for *H. pylori* [8,41,42].

Among Indigenous populations, *H. pylori* infections are seen at greater prevalence for the non-cardia subtype of gastric cancer compared to cardia subtypes [17]. Exact prognosis based on non-cardia versus cardia is inconclusive, but a recent study (2021) indicated that cardia variants may lead to worse outcomes [43]. A cause of concern among Alaska Native individuals is the high rate of antimicrobial resistance, which may affect the treatment efficacy of therapies targeted against *H. pylori* [7,44]. Reinfection has also been found to occur among Alaska Natives after treatment [7,44]. Smoking has also been found to decrease antibiotic treatment efficacy of *H. pylori* among the Indigenous population of Taiwan [4].

## 7. The Health Care System

Lack of access to health care and cultural, geospatial, and attitudinal barriers in the medical system are factors which American Indian and Alaska Natives encounter more frequently when compared to Non-Hispanic Whites [45]. Factors related to cancer prevention, screening, treatment and survival for gastric and other cancers among Indigenous populations are displayed in Figure 2 and discussed below.

### 7.1. Access to Cancer Prevention and Control for All Cancer Types among Indigenous Populations: Challenges of Time, Geography and Communication

Structural inequality can be seen at oncology healthcare centers, which impacts the care of Indigenous cancer patients. Time is a major barrier for Indigenous patients with any type of cancer. American Indian and Alaska Native cancer patients experience longer wait times for initiation of cancer treatment from time of diagnosis, and initiate treatment at lower rates when compared to Non-Hispanic White populations [46]. When treatment is initiated, there is an average ten-week delay among American Indian and Alaska Native patients receiving their first treatment [46]. Long clinic wait times for cancer care and lack of available cancer providers at local health care facilities contribute to lack of treatment and surgical noncompliance among American Indian and Alaska Native patients [48].

Regarding choice of cancer treatment, patients report feeling coerced to proceed with unfamiliar treatments that were not clearly explained, if at all, and others receive treatment without being educated on the consequences or purpose of the treatment [47]. Additional challenges that have been reported among American Indian and Alaska Native populations include access to cancer care facilities, high out of pocket costs, lack of transportation to clinical appointments, and a mistrust of healthcare facilities [46]. As location is a factor in access to food and environmental issues as discussed earlier, it also matters when it comes to accessing health care for Indigenous populations. In Alaska, single jet engine airplanes have been found to be the only form of transportation to healthcare facilities for cancer care treatment for patients from some Alaska Native communities [7,44].

### 7.2. Indigenous Gastric Cancer Patients: Challenges of Access to Cancer Prevention and Control

The published literature has captured limited specific factors related to gastric cancer among Indigenous populations (Figure 2). Frustration with the western medical system has been an ongoing issue. This is exacerbated by inadequate information on prevention and management for gastric cancer, and inconclusive diagnosis requiring additional travel (due to geography) and wait times [28].

As described above for Indigenous patients with all types of cancer, issues surrounding surgical compliance and treatment abound for gastric cancer patients. Surgical noncompliance among American Indian/Alaska Native patients with gastric cancer is high [48]. American Indian and Alaska Native patients begin gastric cancer therapy at a later stage when compared to non-Hispanic Whites [48]. Nonsurgical compliance is an important issue as it leads to worse gastric patient outcomes [48].

## 8. Education and Awareness

### 8.1. Examples of Successful Indigenous Gastric Cancer Programming

While the significance of *H. pylori* in gastric cancer has been established, a lack of understanding of what *H. pylori* is and how it relates to gastric cancer continues to exist in American Indian/Alaskan Native populations [28]. A focus group of 31 Diné members expressed that they had never heard of gastric cancer or *H. pylori*, and they developed great concern upon explanation of the disease [28]. There is no known way to completely prevent gastric cancer, but there are methods and programs for risk reduction. The Navajo Healthy Stomach Project is an initiative meant to better understand how *H. pylori* infections impact the Navajo (Diné) population. This project has shown that additional educational resources and improved public health messaging is desired by the Diné people, a Nation that faces *H. pylori* rates three to four times higher than those of the non-Hispanic white population [28]. A multiagency workgroup in Alaska coordinated a symposium for public health officials and medical staff. This highlighted the problem, focusing on enhanced awareness, and educational outreach, and offered potential solutions for improving gastric cancer prevention and control in Alaska Native populations [7]. The symposium concluded with recommendations, including promotion of improved community education, increased efforts to screen targeted populations, and enhanced surveillance [7].

### 8.2. Inceasing Education: A Potential Tool for Explaining Gastric Biology/Cancer through a Cultural Lens

Colonial medicine perspectives provide the foundation for how western providers and healthcare staff interacts with patients. Culturally based perspectives that encompass Indigenous values and knowledge are needed. The medicine wheel is a representation of harmony and balance and can be applied as a framework for understanding health. Many medicine wheels illustrate the life cycle (children, adolescents, adults, and elders), the four components of well-being (physical, emotional, mental, and spiritual), and other concepts, including fluidity and parts of the whole (direction, season, the elements) [50]. With cancer, the body experiences a state of dysregulation and imbalance [2]. The medicine wheel can be used to map out normal organ function and connect the component functions across levels to visually depict the biologic interplay that leads to health (harmony) or disease (imbalance) [50].

To apply the medicine wheel to gastric cancer, it is important first to understand the purpose of the stomach and how it naturally operates. Four key functions of the stomach are gastrointestinal (GI) motility, enzyme secretion, acid secretion, and its role as a reservoir [51]. Normal function involves each component working together in harmony, and therefore lends itself to a novel medicine wheel that combines both the life cycle and four components of well-being (Figure 3): 1. Physical/GI motility can be categorized by the physical portion of the wheel as it produces the physical movement of products. 2. Emotional/Enzyme activity is rapid and has many moving dynamic states. 3. Mental/Acid secretion plays a protective role by guarding against foreign pathogens [51] and responding to incoming information intelligently. 4. Spiritual/Reservoir contains the accumulation of experience, life lessons and nourishment. This is not meant to be a rigid explanatory model but a starting tool that can be adapted to best fit the needs and cultural values of each tribe or group, grounding their understanding in a noncolonial perspective.

## 9. Discussion

Indigenous populations worldwide are burdened with increased incidence and mortality from gastric cancer due to structural inequalities and lack of cultural responsiveness. We have shown that there is sparse published literature on this population for this type of cancer, with the majority of available research comprised of secondary analyses and or emphasizing biological findings. The majority of findings are limited to those from primarily developed countries and those published in English, a limitation of this study.

The environmental health disparities set in motion by colonialism and resource dispossession play a major role in contributing to increased rates of gastric cancer. Their effect is seen through lack of access to fresh and healthy foods, uranium exposure, and *H. pylori*. The high prevalence of *H. pylori* among Indigenous populations is of concern and risk factors for acquiring *H. pylori* need to be examined further. Measures that would address this are promotion of proper dental care to prevent periodontitis, which is associated with *H. pylori* infection, and improved access to clean water [52]. In addition, programs for clean water access should be developed [7] and policies focused on water regulation and prevention of contamination from environmental pollutants are needed [28].

Changes in diet to less processed, lower sodium foods can decrease gastric cancer risk. The Traditional Foods Resource Guide created by Southeast Alaska Regional Health Consortium includes information on encouraging transitioning diets from highly processed foods to traditional foods [53]. Tribal specific programs such as Utah Diné Bikéyah’s Traditional Foods Program [54], and Indigenous programs such as Indigikitchen conduct cooking classes and demonstrations on how to prepare traditional meals that exclusively use Indigenous ingredients [55]. Resources from these organizations show ways in which Indigenous organizations can help promote awareness of gastric cancer through traditional food resources. Improvements in access to healthy foods may help decrease risk for gastric cancer. Many Indigenous stakeholders are advocating for improved access to healthy foods in schools [56], and conducting research to change how food is produced in Canadian Indigenous settings [13]. Programs that encourage and promote food sovereignty for Indigenous peoples already exist globally, including in Canada [19], Australia [57] and the United States [58]. A strong step to reduce gastric cancer would be for healthy eating programs run by tribal nations and Indigenous organizations to include information on how adjusting diets to include more whole foods and losing weight decreases the risk for developing gastric cancer [11]. Food programs to educate and promote farming practices at home, decrease salt and sugar consumption, exercise, and self-care could also make an impact [28].

From screening to treatment for gastric cancer, there are numerous challenges that relate to education, cost, long travel distances, lack of resources at healthcare facilities and lack of cultural responsiveness. Increased awareness around enhancing screening and compliance with gastric cancer therapy are needed [59]. After an individual receives a cancer diagnosis, the challenges intensify for reasons such as inability to pay for treatment costs, inadequacy of resources, including available physicians, and long travel distances to receive care. To reduce the financial burden, increasing Medicare coverage as well as financial assistance programs to assist with Medicare enrollment or insurance copayments or coverage gaps would be beneficial [45]. Additionally, incorporating telehealth appointments for oncology patients when appropriate would decrease the barrier of travel distance and time [49]. Solutions to improve cancer treatment initiation and wait time for American Indian and Alaska Native populations would entail increased access to cancer care providers and clinics on tribal lands and in rural areas [45]. Patient navigators to help patients traverse the cancer treatment process can potentially decrease barriers by assisting patients with obtaining medical referrals, scheduling appointments, and transporting and accompanying cancer patients during their scheduled appointments [60]. It is also preferable that navigators be American Indian, as American Indian cancer patients prefer Native Patient navigators for emotional support, cultural support, and cancer care coordination [60].

Incorporating culture into all aspects of cancer care from screening to treatment is needed. Figure 3 is a visual representation that may be useful in explaining gastric function and cancer in a culturally appropriate manner to newly diagnosed patients. To reduce the incidence of gastric cancer among American Indian and Alaska Natives, further efforts to develop culturally sensitive cancer screening initiatives, talking circles or discussions, and educational programs about the significance of cancer screening and reduction of lifestyle risk factors would be beneficial for American Indian and Alaska Natives [45]. Providing resources that are culturally responsive for the comprehensive and proactive education of Indigenous communities can allow patients to make the best decisions on the treatment and management of gastric cancer [28].

Some organizations and tribes are working to inform populations about risk reduction and the benefits of screening for gastric cancer, which are important efforts aimed at reducing the structural inequalities mentioned previously. At present, the major Indigenous cancer organizations in the United States do not have webpages for individuals or gastric cancer education materials for reducing risk or promoting screening. Organizations such as the American Indian Cancer Foundation, among others, have culturally tailored resources for breast, cervical, and colorectal cancer with infographics, toolkits, and other programs [59], all of which are excellent and important tools. Any expansion efforts to include these resources on gastric cancer would be valued. Lastly, colonial perspectives that shadow patient interactions and systems delivery interfere with Indigenous communities’ ability to prevent and control gastric cancer on their own terms.

## 10. Conclusions

Numerous factors, primarily stemming from the continued effects of colonialism leading to structural inequalities, have led to Indigenous populations worldwide being burdened with high gastric cancer incidence and mortality. Removing Indigenous people from their homelands to less desirable land without the same infrastructure as that in urban areas has increased food insecurity, contamination of water, water insecurity, and. *H. pylori* infections. It is critical to raise awareness of these inequities so structural changes can be implemented and alleviate the root causes of gastric cancer pathogenesis.

## Figures and Tables

**Figure 1 ijerph-19-05437-f001:**
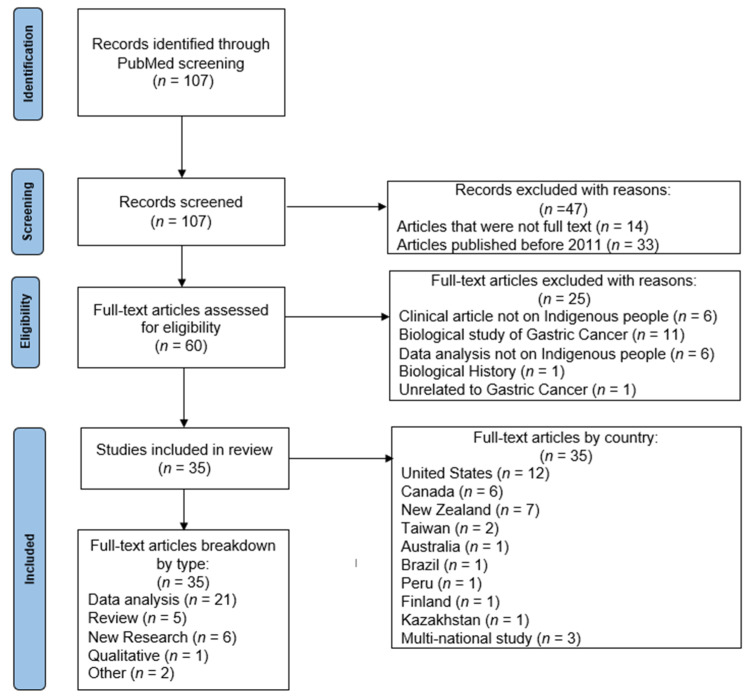
Systematic literature review, PubMed from 2011–2021.

**Figure 2 ijerph-19-05437-f002:**
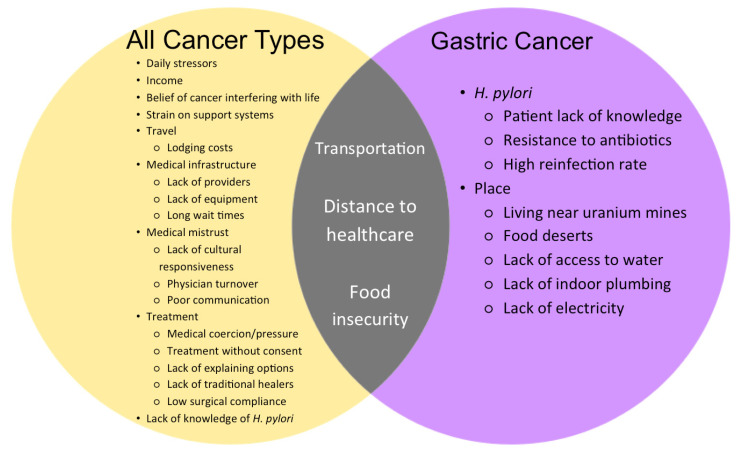
Issues Indigenous patients face in regards to access to cancer prevention and control for all types of cancer (yellow), gastric cancer alone (purple) and both (grey) [7,13,15,16,17,21,23,26,27,28,37,44,45,46,47,48,49].

**Figure 3 ijerph-19-05437-f003:**
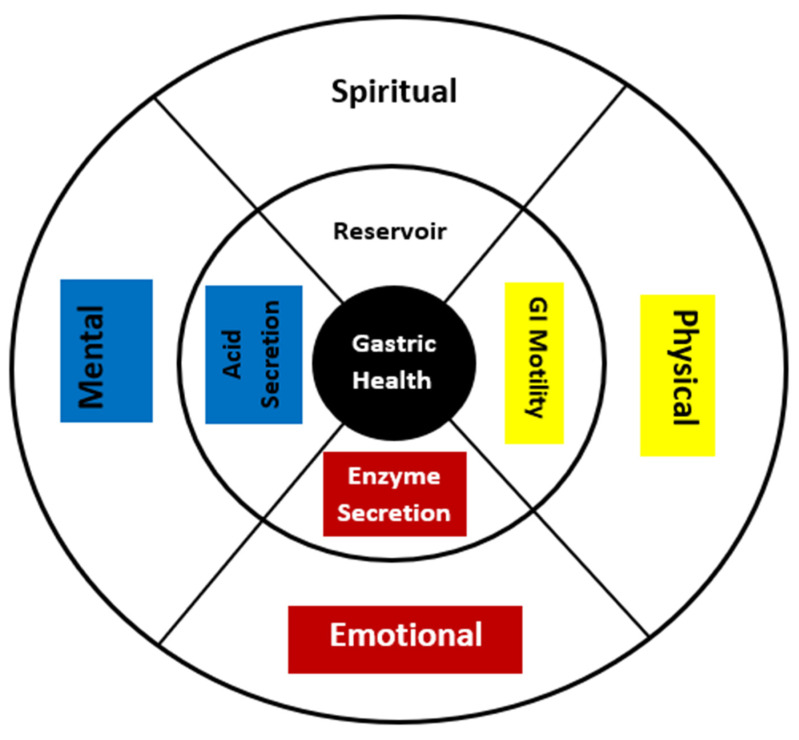
Gastric medicine wheel: combining biology and culture to explain gastric health.

## Data Availability

Not applicable.

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
