# Peer review of "Native and Indigenous Populations and Gastric Cancer: A Worldwide Review"

_ijerph, 2022, doi:10.3390/ijerph19095437_

Round 1

Reviewer 1 Report

The authors have made significant changes and they look appropriate

Reviewer 2 Report

line 118

60.0%%

Please correct the duplication.

This manuscript is a resubmission of an earlier submission. The following is a list of the peer review reports and author responses from that submission.

Round 1

Reviewer 1 Report

  • Summary: The paper focuses on review of the literature on gastric cancer among Indigenous populations globally and identifies unique risk factors, barriers and multiple determinants of health. The strengths include its focus on specific cultural aspects, influence of colonialism and how it impacts the health among Indigenous populations primarily in American Indians and Alaska Natives. The extensive descriptions of the Indigenous populations challenges and possible solutions is also a strength.
  • General concept comments

    The topic of gastric cancer among Indigenous populations is of great significance and there is a need to have a better understanding of the multiple complex factors that could be playing a role.

  • However, the authors presentation of the topic was not very clear and confusing. The authors have presented many different ideas that were put together simultaneously that made it challenging to focus. The same factors were used as risk factors and then as barriers and then as challenges. Terminologies used could be consistent such environmental factors, environmental determinants, environmental violence, environmental racism, it was not clear if they are different or they are same that are used interchangeably. 

  •  

    The use of food as a factor- whether choices or access could be consolidated

  • There is clearly a gap in our knowledge and understanding of all the causal factors for this high burden of disease in Indigenous populations compared to general populations. The Review does cover some aspects of the major risk factors for Gastric cancer, there are many biological and health care related factors that could have been covered more.  However the presentation and the categories were not presented distinctly. It is not presented in a well-structured manner. The sentences were long and had multiple thoughts in one sentence and did not make a case for the category being discussed. Several concepts can be merged into a few major categories. Barriers and risk factors were used interchangeably. All the figures did not add to the main point of discussion, may be a few.

The review is relevant to the field but poorly written and not clear, or comprehensive.

It does make a strong case on the gap in information, and policies that could be considered.

The statements and conclusions drawn  are not coherent and many of them seem to be opinions that are not supported by evidence- such as use of Food as medicine.

The figures could be improved to address the complexity of the determinants.

Author Response

Thank you for your thoughtful comments, we are extremely appreciative for the feedback and the opportunity to strengthen our work.

Please see our responses in bold below, thank you!

Organization

  • However, the authors presentation of the topic was not very clear and confusing. The authors have presented many different ideas that were put together simultaneously that made it challenging to focus.
  • The review is relevant to the field but poorly written and not clear, or comprehensive.
  • Several concepts can be merged into a few major categories. Barriers and risk factors were used interchangeably.
  • The use of food as a factor- whether choices or access could be consolidated

Thank you for bringing this to our attention, we have restructured the manuscript. We have also heavily edited our manuscript for grammar and writing style and verified with our co-author that has a degree in English.

Organization and Writing style:

  • There is clearly a gap in our knowledge and understanding of all the causal factors for this high burden of disease in Indigenous populations compared to general populations. The Review does cover some aspects of the major risk factors for Gastric cancer, there are many biological and health care related factors that could have been covered more.  However the presentation and the categories were not presented distinctly. It is not presented in a well-structured manner. The sentences were long and had multiple thoughts in one sentence and did not make a case for the category being discussed.

This manuscript has been restructured, additional information has been added to the manuscript and writing style and grammar has been edited/verified as being correct. Two of the authors on this manuscript are journal editors and one has a degree in English.

Wording and support

  • The same factors were used as risk factors and then as barriers and then as challenges. Terminologies used could be consistent such environmental factors, environmental determinants, environmental violence, environmental racism, it was not clear if they are different or they are same that are used interchangeably. 
  • The statements and conclusions drawn are not coherent and many of them seem to be opinions that are not supported by evidence- such as use of Food as medicine.

We have restructured and heavily edited the manuscript, information that could more fully support the conclusions have been moved into the conclusion. We have clarified these items.

Figures:

  • All the figures did not add to the main point of discussion, may be a few.
  • The figures could be improved to address the complexity of the determinants.

Figures were presented in a way to add to the main body of the manuscript and not only the discussion. We have changed some of the descriptions of the figures to match the new restructured/edited paper, and believe that this lends itself to highlighting the figures in a more clear manner.

Reviewer 2 Report

 ijerph-1599053 Native and Indigenous Populations and Gastric Cancer: A 2 Worldwide Review

This manuscript describes findings from the literature on Indigenous populations globally and gastric cancer.  This manuscript has major weaknesses requiring major revisions.  This manuscript has the potential to provide synthesized findings on gastric cancer and Indigenous populations globally.  It is recommended that authors review literature review articles to understand how they are published and what is required.

There is no study aim and purpose stated.  Therefore the purpose of this review is not clear at all.  Once the purpose of the study is described, research questions and aims need to be included.  Then all methods and results should align with the purpose and research question.

Methods need to be expanded.  It appears that PRISMA was used.  The particular method used needs to be named.  This needs to be identified with a description and explanation on how it was implemented.  This journals has provided guidelines on describing methodolgoy PRISMA (prisma-statement.org)

Writing needs to be strengthened to academic professional level.  An editor with experience in scientific writing style is recommended

Authors need to consult resources on submitting a literature review manuscript.  Suggestion https://subjectguides.uwaterloo.ca/c.php?g=695509&p=4933476

Author Response

Overall we have heavily edited and restructured the manuscript based on the excellent reviewer comments we received. We thank you for the opportunity to strengthen our manuscript.

Please see our responses in bold below, thank you!

Writing Style:

  • This manuscript has major weaknesses requiring major revisions. 
  • Writing needs to be strengthened to academic professional level.  An editor with experience in scientific writing style is recommended
  • Authors need to consult resources on submitting a literature review manuscript.  Suggestion https://subjectguides.uwaterloo.ca/c.php?g=695509&p=4933476
  • It is recommended that authors review literature review articles to understand how they are published and what is required.

Additional information has been added to the manuscript and writing style and grammar has been edited/verified as being correct. The entire manuscript has been heavily edited and verified with our co-author that has a degree in English and is also an editor for a high impact journal.

Purpose/aim

  • There is no study aim and purpose stated.  Therefore the purpose of this review is not clear at all.  Once the purpose of the study is described, research questions and aims need to be included.  Then all methods and results should align with the purpose and research question.

We have clarified the purpose of the study in the abstract and the introduction

Methods:

  • Methods need to be expanded.  It appears that PRISMA was used.  The particular method used needs to be named. 
  • This needs to be identified with a description and explanation on how it was implemented.  This journals has provided guidelines on describing methodolgy PRISMA (prisma-statement.org)

We have edited the methodology and results section based on your comments, thank you.

Reviewer 3 Report

The present study summarizes the risk factors for gastric cancer in indigenous peoples by conducting a qualitative systematic review. It would be nice to state in the abstract that a qualitative systematic review was conducted. In other words, make it clear that it is not a quantitative systematic review.

The literature search period was 10 years. Why did you make it 10 years? Wouldn't it be a more comprehensive search for 20 years?

The literature adopted in the present study is mostly from developed countries. It is unlikely that such studies will be published by developing countries. I think there may be a bias here. Why not mention the bias that is occurring in the present study in the discussion section?

Line 46
3.4/100,00 for females 
One zero is missing.

Line 228
purchased groceries at gas stations
What does this mean? Does this mean that only processed foods are sold at such stores? Do these stores not sell fresh foods?

Author Response

Overall we have taken in all the comments and have done major editing and restructuring of the manuscript. Thank you for the opportunity to strengthen our work.

Please see our responses in bold below. thank you!

  • The present study summarizes the risk factors for gastric cancer in indigenous peoples by conducting a qualitative systematic review.
  • It would be nice to state in the abstract that a qualitative systematic review was conducted. In other words, make it clear that it is not a quantitative systematic review.

We have added this to the abstract as well as methods, thank you for this comment.

  • The literature search period was 10 years. Why did you make it 10 years? Wouldn't it be a more comprehensive search for 20 years?

There has been another publication that was published in 2014, focusing on up to findings from 2013. We decided to focus on more recent findings as well as to determine if there have been more publications occurring within the past 5 years to see if research in this area has picked up. We have added this information under the methods section.

  • The literature adopted in the present study is mostly from developed countries. It is unlikely that such studies will be published by developing countries. I think there may be a bias here. Why not mention the bias that is occurring in the present study in the discussion section?

In reviewing the other manuscripts published for this special issue, reviews articles and original research articles published did not have a limitations section/did not discuss limitations in their discussion. Even with this, we have added a sentence in the first paragraph of the Discussion stating this as a limitation.

  • Line 46
    4/100,00 for females 
    One zero is missing.

Thank you for bringing this to our attention, we have corrected this.

  • Line 228
    purchased groceries at gas stations
    What does this mean? Does this mean that only processed foods are sold at such stores? Do these stores not sell fresh foods?

We have clarified this statement in the manuscript and added supporting information and a citation. Thank you.

Round 2

Reviewer 1 Report

The authors have made significant changes.  The sentence is needed that talks about more research is needed in this topic, so i would suggest to include that sentence in the abstract.

The section on water is okay but combining it with H. pylori is not a good idea. Separating the H. pylori section with its won title would be important.